# WNT Signaling as a Therapeutic Target for Glioblastoma

**DOI:** 10.3390/ijms22168428

**Published:** 2021-08-05

**Authors:** Michael Latour, Nam-Gu Her, Santosh Kesari, Elmar Nurmemmedov

**Affiliations:** 1Saint John’s Cancer Institute at Providence Saint John’s Health Center, Santa Monica, CA 90404, USA; michael.latour@providence.org (M.L.); santosh.kesari@providence.org (S.K.); 2Samsung Medical Center, Seoul 135-710, Korea; nher@korea.ac.kr

**Keywords:** glioblastoma, WNT, beta-catenin, therapeutic, drug, resistance

## Abstract

The WNT (Wingless/Integrated) signaling pathway is implicated in various stages of glioblastoma, which is an aggressive brain tumor for which therapeutic options are limited. WNT has been recognized as a hallmark of therapeutic challenge due to its context-dependent role and critical function in healthy tissue homeostasis. In this review, we deeply scrutinize the WNT signaling pathway and its involvement in the genesis of glioblastoma as well as its acquired therapy resistance. We also provide an analysis of the WNT pathway in terms of its therapeutic importance in addition to an overview of the current targeted therapies under clinical investigation.

## 1. Biology of WNT Signaling

WNT (Wingless/Integrated) signaling regulates key cellular events during the development of the central nervous system. Particularly, it regulates self-renewal, differentiation, migration and signaling of neural stem cells in the fetal ventricular zone, the postnatal subventricular zone and the hippocampus [1,2]. It has been abundantly demonstrated that hyperactivation of WNT signaling is associated with driving malignant transformation and development of brain tumors [1,3,4]. The level of β-catenin expression is directly linked with the proliferation of neural stem cells and, thus, establishing its importance in the self-renewal and proliferation of these cells [5,6]. 

WNT signaling operates through a group of signal transduction pathways that pass signals from outside of the cell through cell surface receptors into the cytoplasm and then into the nucleus [7]. Three WNT signaling pathways have been characterized: the canonical WNT pathway, the non-canonical planar cell polarity pathway and the non-canonical WNT/calcium pathway [8]. The canonical WNT/β-catenin signaling pathway regulates stem cell self-renewal, cell proliferation and cell-fate decisions of neural stem cells (NSCs) [9]. The β-catenin transcriptional co-activator governs the canonical WNT response; therefore, its distribution and levels within a cell are tightly regulated [10]. When cells such as stem/progenitor cells in the adult hippocampus are not exposed to WNT ligands, cytoplasmic β-catenin associates with a multi-protein “destruction complex” [11]. This complex contains adenomatous polyposis coli (APC), the axis inhibition proteins 1 and/or 2 (AXIN1/2), casein kinase 1 (CK1) and glycogen synthase kinase 3 beta (GSK3β). Here, APC and AXIN1/2 function as scaffolds to position β-catenin in the proximity of CK1 and GSK3β. Phosphorylation of β-catenin by CK1 and GSK3β prime it for ubiquitination by the β-transducin-repeat-containing protein (β-TrCP) and subsequent degradation via the proteasome [12]. In the absence of WNT, the members of the T-cell factor/Lymphoid enhancer factor (TCF/LEF) family of sequence-specific transcription factors bound to WNT responsive DNA elements (WREs) recruit transcriptional corepressor complexes. These complexes include transducin-like enhancer (TLE) and C-terminal binding protein, which associate with histone deacetylases to repress target gene expression [13]. WNT ligands bind to frizzled (FZD)/low-density lipoprotein receptor-related protein (LRP) 5 or 6 receptor complexes expressed on the cell surface. This binding results in the subsequent recruitment of AXIN1/2 to the plasma membrane via interaction with disheveled proteins (DVLs) and inactivation of the destruction complex. β-Catenin then escapes proteasomal degradation, accumulates in the cytoplasm and translocates into the nucleus where it displaces TLE from TCF/LEF bound WREs [13]. β-Catenin/TCF/LEF complexes, in turn, recruit histone-modifying complexes, such as CREB-binding protein (CBP)/p300 protein acetyltransferases, mixed lineage leukemia (MLL)/Set methyltransferases and chromatin remodeling complexes including switch/sucrose nonfermentable (SWI/SNF) to induce the expression of WNT/β-catenin target genes and drive cellular proliferation [14].

The non-canonical planar cell polarity pathway is initiated by the binding of WNT proteins to FZD; the signal is then transduced to DVL, which subsequently activates the GTPases Rho and Rac. Rac then activates c-Jun N-terminal kinase-dependent transcription and Rho activates Yes-associated protein/Transcriptional Co-activator with PDZ-biding Motif-dependent transcription [15]. Since the planar cell polarity pathway regulates cytoskeletal rearrangement and cell movement, the pathway is heavily utilized during embryogenesis and the invasion/metastasis of cancer cells [16]. In the non-canonical WNT/calcium pathway, bound FZD receptors activates DVL and phospholipase C (PLC), which subsequently produces the 1,4,5-triphosphate (IP3) that binds to and opens calcium channels found on the endoplasmic reticulum membrane [17]. NFAT1, which is a Ca^2+^-dependent transcription factor that is down stream of this pathway, is highly expressed in GBM where it regulates invasion [18].

Recent efforts have been spent on targeting the β-catenin/TCF4 complex, i.e., the catenin responsive transcription (CRT), with an expectation to inhibit its downstream signaling. The CRT complex interacts with several gene regulators through a conserved mechanism, thus raising questions about the specificity of this approach. One recent study reports the discovery of inhibitor iCRT-3, which targets the CRT complex [19]. While this provides hope for direct targeting of β-catenin, it is not clear whether they trigger the degradation of β-catenin, thus raising concerns about the possibility of its re-localization back into the nucleus. The mechanism of action of these compounds needs to be further elucidated to assess suitability for therapeutic use. A more specific inhibitor, stapled peptide StAx35R, is reported to block AXIN from binding to β-catenin [20]. While the peptide is large enough to cover a significant surface area on the target, it carries the drawback of low cell penetration, thus necessitating its use at high doses. The clinical advantage of the stapled peptide class of molecules has yet to be established [20]. More recent studies focused on the identification of direct small-molecule inhibitors of β-catenin have results in the discovery of C2 and MSAB, which are demonstrated to sequester β-catenin and cause its degradation via the ubiquitin degradation system [21,22].

Glioblastoma multiforme (GBM) is recognized as the most aggressive primary malignant brain tumor with a median survival of only 12–18 months upon diagnosis [23]. The standard of care for GBM has not advanced much recently; it includes surgical resection of the tumor followed by radiation therapy and combined with temozolomide (TMZ) chemotherapy [24]. As a DNA-methylating agent, TMZ is the most used chemotherapy to treat GBM. Despite the recent advances in GBM biology, the average survival rate of GBM patients has not improved significantly. Even though these limited therapeutic options provide temporary relief, the course of GBM eventually results in relapse and acquired resistance, which remain a major clinical challenge [24,25]. Hence, a detailed understanding of the molecular mechanisms of GBM resistance and the interplay of neural stem cells is of paramount importance for the development of novel therapeutic approaches as well as personalized medicine regimens.

GBM tumors display a high degree of complexity in terms of heterogeneity and cellular stratification—this indicates genomic and molecular diversity profiles of tumors [26,27,28]. Proneural and mesenchymal subtypes are two leading GBM subtypes identified through multiple studies [29,30,31]. On the level of cellular stratification, there are subpopulations with distinctive features enabling tumor initiation and propagation, as well as tumor growth and treatment resistance [32,33,34,35]. What is of paramount importance is the role of WNT signaling in these cell events and hierarchies as it promotes GBM growth and invasion and therapy resistance [36,37,38,39].

## 2. WNT Signaling in GBM Pathogenesis and Progression

### 2.1. The Role of WNT in Gliomagenesis

Despite accounting for less than 1% of cells within any given GBM tumor, cancer stem cells known as glioma stem cells (GSCs) have been associated with the origin and recurrence of GBM [40]. These tumorigenic glioma stem cells share various characteristics with normal neural stem cells (NSCs), including self-renewal and the capacity to modulate between proliferative and quiescent states [41,42]. NSC niches, especially the subventricular zone (SVZ), have been associated with the population of cells that give rise to this primary malignant tumor. In a study focused on IDH-wild-type GBM, 56% of screened patients carried shared low-level driver mutations between tumor-free SVZs and their matched tumor tissue, which suggests that the NSCs harboring somatic mutations can migrate from the SVZ and transform into GSCs [43]. In addition to NSCs, NSC-derived astrocytes and oligodendrocyte precursor cells are also candidates for the cells of origin in GBM [4]. Since GSCs have the capacity to recreate a whole tumor in xenograft assays and produce heterogeneous populations of cells with varying degrees of differentiation, a great deal of GBM research has been focused on interrogating the dysregulated signaling pathways that maintain the stem cell-like phenotype of GSCs [44].

The WNT pathway, along with Notch, Hedgehog, RAS/RAF/MAPK and PI3K/Akt/MTOR, constitute the main signaling transduction pathways that confer stemness in GBM [45]. NSC development requires WNT signaling [46]. The WNT system is often overactive in GBM, allowing cells to recapitulate embryonic processes that result in the characteristic proliferation and invasiveness seen in GBM tumors. While mutations relative to the WNT pathway, except for the homozygous mutation of FAT apical cadherin (FAT1), are considered rare in GBM, epigenetic alterations to the pathway often occur in a manner that represses WNT inhibitors and promotes β-catenin activity [47,48]. Expression levels of canonical WNT factors such as WNT3A and TCF4 are positively correlated with glioma grade and LEF1, which is another canonical WNT transcription factor, is associated with poor clinical outcomes [49,50]. HOXA13, a homeobox gene, activates genes associated with the WNT signaling and promotes GBM development via activation of the canonical WNT pathway [51]. WNT/β-catenin signaling has been shown to transcriptionally regulate vascular endothelial growth factor (VEGF), a key mediator of angiogenesis; The VEGF gene promotor has been found to have seven TCF binding sites [52]. The activation of angiogenesis demonstrates yet another WNT-induced mechanism that supports GBM growth. As a pathway that regulates the stemness of NSC, the aberrant activation of the canonical WNT cascade has been associated with various cancers, including gliomas [46,53]. However, the entire WNT system, including the non-canonical WNT signaling, has been implicated in the role of gliomagenesis.

WNT5A, a non-canonical WNT molecule, increases neural differentiation and is expressed during NSC proliferation [54,55]. ShRNA knockdown of WNT5A in GBM-05 and U76MG resulted in the reduction in the proliferation in these GBM cells [55]. The activation of the non-canonical WNT signaling pathways is more associated with the invasiveness of GBM. Expression levels of WNT5A and Frizzled-2 (FZD-2), another non-canonical WNT factor, are correlated with GBM invasiveness [56]. In addition to being associated with a worse prognosis, high WNT5A expression levels can be used to delineate between some GBM subtypes in the TCGA dataset: WNT5A expression is higher in mesenchymal GBM as compared with classical GBM [57]. Additionally, mesenchymal human GBM tissue has been observed to produce a widespread signal of WNT5A immunoreactivity upon IHC analyses, whereas proneural and classical tissue samples only had a few WNT5A-positive cells [58]. WNT5A binds to tyrosine kinase-like orphan receptor (ROR) 1 or (ROR2) and FZD, resulting in receptor internalization and the initiation of the PCP pathway [59]. JNK, which is a downstream factor of the PCP pathway, facilitates WNT5A-induced formation of lamellipodia and reorientation of the microtubule-organizing center (MTOC) [60]. WNT5A knockdown in U251 cells results in a reduction in migration during would healing, suggesting that WNT5A regulates the cell motility of GBM cells [61]. Due to the upregulation of non-canonical WNT pathways, mesenchymal tumors are more motile and invasive than the GBM tumors that are classified as proneural or classical [62].

The canonical WNT pathway and the non-canonical (β-catenin-independent) WNT pathways are involved in promoting components of the epithelial-to-mesenchymal transition (EMT), a complex process that describes a pattern of phenotypic changes observed during embryonic development and regeneration. The EMT program, which has a role in metastasis, is hardly ever fully executed in solid tumor cells. Executing the endpoint of the transition would presumably result in differentiation, which generally follows the EMT events that occur during embryogenesis, and therefore a reduction in adaptive potential [35,40]. EMT-activating transcription factors (EMT-TFs) are currently the most important components of the classical EMT that is relevant to cancer biology [63]. Although metastasis is rare in GBM, the brain malignancy is defined as a grade IV glioma for being highly invasive [64]. Unsurprisingly, WNT signaling is partially involved in mediating the partial EMT observed in various cancer types [63]. WNT/ β-catenin pathway activation results in the upregulation of EMT-TFs such as Twist, Snail, Slug and Zeb1 [29,65,66]. Additionally, increased cell motility and increased Zeb1 expression have been observed in GBM cells with constitutively active beta-catenin [29].

As previously stated, the non-canonical signaling predominantly activates the WNT/Ca^2+^ and planar cell polarity pathways [67,68]. These pathways are associated with processes that result in cytoskeletal rearrangements, which is consistent with the observation that cell migration involved in EMT requires constant reorganization of the cytoskeleton [46,69]. The noncanonical WNT ligands WNT5A and WNT11, which activate these pathways, induce cell migration in Xenopus and zebrafish embryonic models [70]. The utilization of the noncanonical pathway in many GBM tumors is further supported by expression levels of WNT inhibitory factor 1 (WIF1), which is a secreted WNT signaling agonist. In addition to being an inhibitor of the canonical WNT pathway, WIF1 is also able to selectively attenuate the WNT/Ca^2+^ pathway [71]. In a study looking at the gene expression profiles of 80 human GBM samples, 75% of GBM tumors downregulated WIF1. Furthermore, WIF1 deletion or hypermethylation occurred in 10% and 26% of samples, respectively [72].

The capacity for aberrant WNT signaling to initiate and maintain stem-like characteristics in a subset of GBM tumor cells implicates this system in the onset, progression and recurrence of GBM, although the extent to which it is involved in each stage is highly variable among patients and within tumors. Although tumorigenic cells from GBM tissue will be enriched with WNT-dependent GSCs, subpopulations of GSC that are not functionally reliant on WNT signaling have been observed in GBM samples. Nevertheless, WNT-dependent GSCs are a pharmacologically relevant subpopulation of cells, as they are more associated with an aggressive phenotype [73]. WNT signaling maintains stemness in part by upregulating positive modulators of the partial EMT program, particularly the components that would normally create a transient dedifferentiated state. In addition to influencing cell morphology and cell migration, EMT-TFs can facilitate the initiation and/or maintenance of cancer stem cells [63]. WNT/beta-catenin pathway activity upregulates the expression of c-MYC, which, when overexpressed, results in EMT activation in mammary epithelial cells [74]. Additionally, NFAT activation, which occurs in non-canonical WNT signaling, promotes EMT in mouse embryonic stem cells [75]. Furthermore, a study on pancreatic ductal adenocarcinoma provided mechanistic evidence for the regulation of Snail and Zeb1, which are established EMT-TFs, by NFATc1 and further cements the role of both arms of the WNT system in the partial EMT phenotype. GSCs exploit various components of developmental signaling networks, with the EMT being one of many [76].

The importance of viewing the WNT system as a highly conserved means of cellular communication cannot be understated. The implications of an aberrant communication system extend at the very least to non-tumor cells in the tumor microenvironment, which is a crucial factor in cancer progression. Exposing microglial cells, which are common in the tumor microenvironment, to GBM-conditioned medium results in an increase in WNT3A expression and cell viability [76]. GBM-derived WNT3A stimulates the induction of a M2-like phenotype on microglial cells. This is consistent with the observation that the mesenchymal GBM subtype, which is characterized by the upregulation of angiogenesis, is associated with the increased presence of tumor-associated microglia [77,78]. The two-way interactions—mediated by WNT signaling—between the microenvironment and tumor cells add further complexity to GBM pathology. 

### 2.2. WNT Signaling in TMZ Resistance and Radioresistance

WNT signaling has a role in many of the most pernicious characteristics of GBM, including resistance to therapeutics. Similar to GBM onset and progression, the inevitable recurrence after stand-of-care treatment is thought to be primarily mediated by GSCs. Cancer Stem cells CSCs, as well as normal stem cells, naturally possess an increased capacity for DNA repair and predisposes them to preferentially survive therapeutics that target DNA [79]. Another cytoprotective mechanism used by cancer stem cells is the increased expression of a subclass of ATP-binding cassette (ABC) transporters, which are efflux pumps that non-specifically remove drugs within cells [80,81]. ABCB1 is an efflux pump that is expressed in 70–100% of high-grade gliomas [82]. The expression of this drug efflux pump is partially regulated by the non-canonical WNT5A ligand [83]. Modulation between proliferative or quiescence states in NSCs is a highly regulated process. Cdc42, a Rho-GTPase and downstream target of the non-canonical WNT pathway, helps maintain the quiescent state of NSCs in the SVZ [84]. For CSCs, quiescence can operate as a survival strategy to protect against cytotoxic conditions [85]. Single cell transcriptomic analysis of GBM tumors has revealed subpopulations of quiescent GSCs [86]. Consequently, radiation and chemotherapy harms proliferating tumor cells, which constitute the majority of any bulk tumor and inadvertently enriches CD133^+^ GSCs [87]. The WNT system is implicated in TMZ resistance and radioresistance by virtue of maintaining stemness, which, as mentioned above, intrinsically equips GSCs with the machinery needed to survive GBM treatment.

The methylation status of O6-methylguanine-DNA methyltransferase (MGMT) is perhaps the most important epigenetic marker in the context of GBM treatment options. One of the predominant TMZ-induced DNA adducts, O6-methylguanine, is repaired by this mismatch repair protein [88,89]. TOPFlash reporter assay experiments have demonstrated that TMZ treatment results in an increase in WNT signaling activity [90]. Investigations into the relationship between the canonical WNT pathway and MGMT gene regulation revealed that β-catenin knockdown in LS174T cells results in a decrease in MGMT expression. However, the WNT system mediates therapy resistance in various manners that extend beyond upregulating a repair protein.

Autophagy can operate as a pro-survival mechanism that assists in the TMZ resistance of GBM cell lines [91]. ATG9B, which, according to the TCGA database, is significantly upregulated in GBM compared with low-grade gliomas, is an autophagy-related (ATG) protein that mediates the early phase of autophagy [92,93]. The result of blockage of the canonical WNT pathway in LN229 cells was a suppression of ATG9B expression, which is considered to be critical for TMZ-induced autophagy [92]. WNT activity also mediates chemoresistance in the microenvironment; it has recently been shown that GBM chemoresistance is aided by canonical WNT signaling that promotes stem-like activation and mesenchymalization of endothelial cells (EC) [94]. These transformed ECs subsequently upregulate proteins involved in chemoresistance, namely multidrug resistance-associated protein-1 expression.

## 3. WNT Pathway Inhibitors in the Clinic

WNT pathway proteins have been considered an attractive and validated cancer target for its critical roles in tumorigenesis and cancer progression as described above. Over the past decade, much effort has been made to develop WNT pathway inhibitors, some of which have been tested in clinical studies [95,96]. However, there have been no FDA-approved drugs targeting WNT pathways so far. We reviewed and summarized the key drugs that have entered the clinical trials below (Table 1).

### 3.1. Canonical WNT Pathway Inhibitors in Clinical Trials

Vantictumab is a human IgG2 monoclonal antibody that binds Frizzled (FZD) receptors 1, 2, 5, 7 and 8 in the WNT signaling pathway and thereby prevents the activation of canonical WNT signaling [97]. This antibody interacts with FZD receptors through a conserved epitope within the extracellular domain. In a phase 1b dose escalation study in patients with locally recurrent or metastatic HER2-negative breast cancer, multiple patients experienced fractures related to vantictumab which resulted in premature termination [98]. Similarly, a phase 1b study evaluating vantictumab in combination with nab-paclitaxel and gemcitabine in patients with untreated metastatic pancreatic adenocarcinoma was recently closed due to concerns surrounding bone-related safety [99]. The canonical WNT pathway regulates many cellular processes, including bone homeostasis. Ectopic expression of the canonical ligand WNT10B can increase bone mass in transgenic mice [100]. Additionally, WNT16, which is another canonical WNT ligand, has been shown to regulate cortical bone thickness and mineral density [101].

Ipafricept is a recombinant fusion protein comprising the FZD8 cysteine-rich domain (CRD) fused to the human Fc domain [102]. This fusion protein competes with the native FZD8 receptor for its ligands and antagonizes canonical WNT signaling. Ipafricept has shown reduced tumor growth and decreased CSC frequency in preclinical models [103]. However, recent clinical studies in epithelial ovarian cancer patients also resulted in the occurrence of fragility fractures at doses associated with efficacy [104]. While a study in patients with untreated metastatic pancreatic adenocarcinoma showed reasonable tolerance [105], overall safety concerns limit further development of Ipafricept.

The radioimmunoconjugate drug, 90Y-OTSA101, comprises a humanized monoclonal FZD10 antibody OTSA101 labelled with 90Y [106]. Radioactive isotope-conjugated OTSA101 showed strong anti-cancer activity by tumor-specific irradiation in a mouse xenograft model. In a phase 1 clinical study in patients with synovial sarcomas (SS), the best response was stable disease in 3/8 patients lasting up to 21 weeks for 1 patient [107,108].

PORCN is a membrane-bound O-acyltransferase (MBOAT) involved in the posttranslational modification of WNT ligands, which is an important step for WNT ligand secretion. Therefore, the development of small molecule inhibitors that block PORCN have gained increasing attention. WNT974 (LGK974) is a first-in-class PORCN inhibitor that was shown to be potent and efficacious in multiple tumor models [109]. In an ongoing clinical study, biomarker analyses show that WNT974 inhibit WNT pathway activity in tumors [110]. In a recent 2020 AACR conference, a phase I clinical trial of WNT974 demonstrated that combination therapy using WNT974 and the monoclonal PD-1 antibody spartalizumab, was well tolerated in cancer patients with advanced solid tumors in a variety of cancers.

ETC-159 is an orally administered and potent porcupine inhibitor that has shown preclinical efficacy in combinations with phosphoinositide 3-kinase (PI-3K) inhibitors and poly (ADP-ribose) polymerase (PARP) inhibitors [110,111]. ETC-159 was shown to inhibit WNT signaling at doses that were well tolerated in the clinical study [111,112,113]. CGX-1321 is the latest porcupine inhibitor that has entered a clinical trial. Preclinical studies demonstrated that inhibition of the WNT/βcatenin pathway using CGX-1321 combined with paclitaxel reduced tumor size and proliferation in a syngeneic mouse ovarian cancer model [114]. CGX-1321 was also shown to attenuate cardiac hypertrophy in a mouse model and is being evaluated in a phase 1 study in solid tumors [115].

PRI-724 is a first-in-class inhibitor of the interaction between β-catenin and its coactivator CBP. PRI-724 potently disrupts β-catenin interaction with CBP, thereby inhibiting acute myeloid leukemia (AML) cells and synergizing with FLT3 inhibition in *FLT3*-mutant AML [116]. It also showed anti-cancer effects in testicular germ cell tumors and human osteosarcoma cells [117,118]. A recent study revealed that PRI-724 is a potent antifibrotic agent in the lungs [119]. In a phase 1 study, PRI-724 combined with gemcitabine was safe and demonstrated modest clinical activity [120,121]. There are currently no active clinical trials investigating the use of canonical WNT signaling inhibitors to treat patients with GBM. Such a trail would have to find a method to not only bypass the issues of bone toxicity but also the restrictions posed by the blood–brain barrier (BBB), which is the greatest impasse for central nervous system (CNS) targeted drug delivery. Although novel drug delivery systems, such as ligand-anchored dendrimers that utilize receptor-mediated transcytosis to achieve high drug localization [122], may mitigate the systemic toxicity of canonical WNT inhibitors, the side effects of inhibiting the brain activity associated with WNT/beta-catenin signaling, which regulates adult neurogenesis [9], will further complicate treatment strategies. Nevertheless, the evidence on the integral role of canonical WNT signaling in GBM onset and progression is abundant and should be examined in a clinical trial that explores viable drug delivery options.

### 3.2. Noncanonical WNT Pathway Inhibitors in Clinical Trials

In addition to the canonical WNT pathway targets, non-canonical WNT pathways are emerging as a new cancer target. One of the β-catenin-independent WNT pathway targets is a co-receptor for WNT ligands, ROR1 [123]. ROR1, an oncofetal protein important for embryonic development, is expressed in multiple hematologic and solid tumors but not on normal tissues [124]. The overexpression of ROR1 in various types of human cancers has attracted interest from the field of drug discovery [125,126]. Below are several ROR1-targeting therapeutics currently explored in clinical studies.

Cirmtuzumab is a humanized monoclonal antibody that targets ROR1. Treatment with Cirmtuzumab and ibrutinib was shown to be highly effective in clearing leukemia cells in vivo [127]. Moreover, in a phase 1 study involving 26 patients with progressive, relapsed or refractory chronic lymphocytic leukemia (CLL), Cirmtuzumab was safe and effective at inhibiting tumor cell ROR1 signaling in patients [128]. A recent interim phase 1 study evaluating the effects of Cirmtuzumab in combination with ibrutinib showed that the overall best objective response rate (ORR) of patients with mantle cell lymphoma (MCL) was 87% (13 of 15 evaluable patients).

VLS-101 is an antibody-drug conjugate (ADC) for suppressing ROR1-positive cancers. VLS-101 comprises UC-961 (Cirmtuzumab), which is a maleimidocaproyl-valine-citrulline-para-aminobenzoate (mc-vc-PAB) linker, and the anti-microtubule cytotoxin monomethyl auristatin E (MMAE). VLS-101 dramatically decreased tumor burden in all Richter syndrome-colonized tissues and significantly prolonged survival in mouse models [129]. Recently, VelosBio announced the initiation of a phase 2 clinical trial (NCT04504916) to evaluate VLS-101 in patients with solid tumors. In a phase 1 study, VLS-101 resulted in complete responses in 47% (*n* = 7/15) of patients with mantle cell lymphoma (MCL) and 80% (*n* = 4/5) of patients with diffuse large B-cell lymphoma.

Another ADC for ROR1 is NBE-002; NBE-002 has recently entered clinical trials to evaluate its safety and efficacy in patients with triple negative breast cancer and other solid tumors. NBE-002 uses a novel anthracycline payload which has been shown to induce a long-lasting anti-tumor immunity in preclinical models of solid tumors. 

Interestingly, the first chimeric antigen receptor (CAR) T-cell therapeutic for ROR1 developed by BMS/Celgene has entered a clinical trial [130]. Additionally, at least five different ROR-1-centric drugs are being evaluated in preclinical stage [128,130,131,132]. The toxicity associated with targeting this non-canonical WNT receptor is dependent on the expression levels of ROR1 in off-target tissues; ROR1 expression is observed at low levels in adipocytes, which, based on the observed levels of plasma adiponectin following the transfer of autologous ROR1 T-cells into macaques, appear to be lysed at a low-level during CAR T-cell therapy. As the branch of the WNT system more associated with invasion and migration, the non-canonical WNT pathway appears to be a less toxic pharmacological target. Since surgical resection is often incorporated into the treatment of GBM, researchers looking to target this pathway for drug discovery will have to be concerned with the challenges of impairing wound healing, as ROR1 knockdown significantly inhibits cell migration [133]. Each treatment modality, whether it be small molecule or CAR T-cell therapy, will have unique CNS-specific challenges. There are currently no active clinical trials investigating the use of non-canonical WNT signaling inhibitors to treat patients with GBM. As evidenced by the emergence of preclinical and clinical trials investigating ROR1 as a drug target, the non-canonical WNT system appears to be a viable and relatively safe pathway to attenuate in GBM treatment strategies.

## 4. Conclusions

For successful WNT pathway drug development, strategies to reduce normal cell toxicity and selectively kill cancer cells are necessary. Additionally, overcoming the unique challenges of targeting the CNS will require novel improvements to drug modalities and precise clinical trial design. As genomics, epigenomics, proteomics and related fields evolve, new drug targets derived from the discovery of unique molecular characteristics of WNT signaling in GBM will supply researchers with more options for treating this devasting condition. Testing new targets to identify untapped opportunities for WNT pathway therapies may help mitigate the negative impacts on tissue homeostasis and regeneration. Previous WNT pathway inhibitors have largely relied on antibodies and small molecules. Given the recent clinical data on ROR1-ADC, new modalities, such as ADC, CAR-T, CAR-NK and proteolysis targeting chimera (PROTAC), provide promising approaches that will either provide better patient outcomes or at least produce valuable data that will ultimately guide researchers to improved treatment solutions.

## Figures and Tables

**Table 1 ijms-22-08428-t001:** Key WNT pathway inhibitors in clinical trials.

Drugs	Target	Modality	Stage	Identifier	Indication
Ventictumab	FZD1,2,5,7,8	Monoclonal antibody	Phase 1	NCT01973309 NCT01345201 NCT01957007 NCT02005315	Breast, Pancreatic and Solid tumors
Ipafrecept	FZD8 ligands	Recombinant fusion protein	Phase 1	NCT01608867 NCT02069145 NCT02092363	Ovarian, Pancreatic, Hepatocellular and Solid tumors
90Y-OTSA101	FZD10	Antibody-radioactive isotope conjugate	Phase 1	NCT01469975	Synovial sarcomas
WNT974	PORCN	Small molecule	Phase 2	NCT02649530 NCT01351103 NCT02278133 NCT02050178	Head and Neck; Pancreatic, colorectal and Solid tumors
ETC-159	PORCN	Small molecule	Phase 1	NCT02521844	Solid tumors
CGX-1321	PORCN	Small molecule	Phase 1	NCT03507998 NCT02675946	Solid tumors
PRI-724	β-Catenin /CBP	Small molecule	Phase 2	NCT02413853 NCT01764477 NCT01302405 NCT01606579	AML, CML and Solid tumors
Cirmtuzumab	ROR1	Monoclonal antibody	Phase 2	NCT03088878 NCT02776917 NCT02222688 NCT02860676	Breast cancers, Lymphoma
VLS-101	ROR1	Antibody-drug conjugate	Phase 2	NCT03833180 NCT04504916	TNBC, NSCLC and Breast cancers
NBE-002	ROR1	Antibody-drug conjugate	Phase 2	NCT04441099	Advanced solid tumors

## Data Availability

Not applicable.

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
