# Peer review of "WNT Signaling as a Therapeutic Target for Glioblastoma"

_ijms, 2021, doi:10.3390/ijms22168428_

Round 1
Reviewer 1 Report
The paper includes a comprehensive understanding of Wnt pathway in GBM. There are some comments: Part 1: (1. Biology of WNT Signaling) 1. The second and third paragraphs of the "Biology of WNT Signaling" section are about GBM, it might be more reasonable to move to the end of the section, as a link to the "WNT Signaling in GBM Pathogenesis and Progression" section. 2. Lines 45-77: Authors mentioned about 3 WNT signaling pathways: mechanisms that activate WNT signal pathways. Authors described in details about one WNT pathway (canonical WNT pathway). Authors also mentions other pathways related to GBM pathogenesis or therapeutics. Authors might want to add and describe in details about the other two WNT pathways as well. In addition, about the canonical WNT pathway, it seems that the description is related to CRC rather than GBM. Authors may want to summarize and describe the WNT pathway which can be related to GBM, or need to emphasize the mechanism associated with which kind of diseases to avoid confusion. Part 2 (2. WNT signaling in GBM pathogenesis and Progression) 1. Authors point out Wnt5a (line 127-140) maybe a good target for treatment GBM. How about other molecules in Wnt family (such as Wnt3a, Wnt8a, Wnt9b....etc.) ? Does other Wnt molecules have effect in GBM? 2. Does GBM have any mutation in WNT pathway? If yes, the author should discuss it in the second section " WNT Signaling in GBM Pathogenesis and Progression ". 3. Authors may want to add more detail about the pathological mechanism of GBM related WNT signaling pathway including both conventional and unconventional ones (lines 112-199). 4. Lines 200-237: Authors discussed the WNT-related TZM and radioresistance. However, the change of proteins mentioned is not described anywhere, and how it is related to the WNT signaling pathway also needs to be described in more detail. Part 3 (3. WNT Pathway inhibitors in the Clinic) 1. As a reader, I hope to find the information about WNT inhibitors that are currently studying or investigating in clinical in this part. However, the author listed some WNT inhibitor for many diseases, but not including GBM, or if there is no clinical trials, authors may want to give any strategy based on the found target on WNT in GBM (e.g. from part 1). The author only stops in describing the WNT inhibitors or their mechanism without connection to the WNT mechanism on GBM. 2. The section 3-3 (3.3. Challenges and opportunities of WNT pathway inhibitors) could be a new forth section and authors can write more detail.Author Response
attached.

Reviewer 2 Report
This is a review well articulated in all parts and appropriately updated. There are only minor points to revise, which are the following:
- English: there some sparse errors throughout the text needing corrections
- Abbreviations: they should be adequately explained and then used. For instance, cancer stem cells should be abbreviated the first time they appear in the text and thereafter used (see at pag.... lines ...)
- As for Wnt antibody use causing bone damages, the Authors should mention that Wnt pathway is involved in bone morphogenesis and homeostasis. This evidence could explain the constance in bone fragility/damages following the inhibition of this pathway.
